# Diversity of phlebotomine sand flies and molecular detection of trypanosomatids in Brumadinho, Minas Gerais, Brazil

**Aline Tanure, Felipe Dutra Rêgo, Gabriel Barbosa Tonelli, Aldenise Martins Campos, Paloma Helena Fernandes Shimabukuro[ID], Célia Maria Ferreira Gontijo, Gustavo Fontes Paz, José Dilermando Andrade-Filho[ID]***

Grupo de Estudos em Leishmanioses–Instituto René Rachou–Fiocruz Minas–Belo Horizonte, Minas Gerais, Brasil

\* jose.andrade@fiocruz.br

## Abstract

This study aimed to describe the sand fly fauna and detect trypanosomatids in these insects from Casa Branca, state of Minas Gerais, Brazil, an endemic area of both visceral (VL) and tegumentary leishmaniasis (TL). Sand flies were collected bimonthly from May 2013 to July 2014, using automatic light traps exposed for three consecutive nights in peridomiciliary areas of nine houses with previous reports of VL and TL. ITS1-PCR and DNA sequencing were performed for trypanosomatids identification. A total of 16,771 sand flies were collected belonging to 23 species. The most abundant species was *Nyssomyia whitmani* (Antunes & Coutinho, 1939) (70.9%), followed by *Lutzomyia longipalpis* (Lutz & Neiva, 1912) (15.2%) and *Migonemyia migonei* (França, 1920) (9.1%). *Leishmania amazonensis* DNA was detected in *Ny. whitmani* (four pools) and *Le. braziliensis* DNA was detected in *Psychodopygus lloydi* (one pool). In seven pools of *Ny. whitmani* and in one pool of *Lu. longipalpis* positive for *Leishmania* DNA, the parasite species was not determined due to the low quality of the sequences. Moreover, DNA of *Herpetomonas* spp. was detected in *Ny. whitmani* (two pools) and Cortelezzii complex (one pool). DNA of *Crithidia* spp. was detected in *Ny. whitmani* and *Ps. lloydi* (both one pool). Our results suggest that *Ny. whitmani* may be involved in the transmission of *Le. amazonensis* in the study area. The molecular detection of *Le. amazonensis* suggests the presence of this species in a sylvatic cycle between vertebrate and invertebrate hosts in the region of Casa Branca. Our data also reveal the occurrence of other non-*Leishmania* trypanosomatids in sand flies in Casa Branca District.

## Introduction

Phlebotomine sand flies (Diptera: Psychodidae: Phlebotominae) are known to be natural hosts of etiological agents such as bacteria, viruses and protozoa [1,2]. Among these agents, are species of the genus *Leishmania*, which are transmitted during blood feeding by female sand flies [1]. Several sand flies species have been considered proven or suspected vectors of *Leishmania* parasites in Brazil, suggesting that multiple vectors may be involved in parasite cycles, making the epidemiological picture more complex than currently recognized [3,4].

**Data Availability Statement:** Data are available within the manuscript.

**Funding:** JDAF - 302701/2016-8 Conselho Nacional de Desenvolvimento Científico e

Tecnológico (www.cnpq.br) JDAF - PPM-00792-18 Fundação de Amparo à Pesquisa de Minas Gerais (www.fapemig.br) AT - Finance Code 001 Coordenação de Aperfeiçoamento de Pessoal de Nível Superior (www.capes.gov.br). The funders had no role in study design, data collection and analysis, decision to publish, or preparation of the manuscript.

**Competing interests:** The authors have declared that no competing interests exist.

Sand flies are also possible natural vectors of several trypanosome species, both dixenous (i.e. those with two hosts in their life cycle—*Trypanosoma*, *Endotrypanum*, and *Phytomonas*) and monoxenous (i.e. those having only one host—*Leptomonas*, *Crithidia*, *Blastocrithidia* and *Herpetomonas*) [5–10]. However, the identification of these protozoa has been considered irrelevant from the epidemiological perspective. Recent studies have discussed the evolution of trypanosome taxonomy and, despite improvements have been made, the resolution of evolutionary relationships within the Trypanosomatidae is confounded by our incomplete knowledge of its true diversity [11–15]. More recently, several monoxenous flagellates have been identified from human clinical isolates and some of them, similar to both TL and VL [16–21]. Several authors have discussed the capacity of these trypanosomatids to cause leishmaniasis in a possible co-infection with *Leishmania* or even in a single infection [19,20,22,23]. The dixenous parasite *Endotrypanum colombiensis*, until recently *Leishmania colombiensis* [12], also represent another relevant Trypanosomatidae related to leishmaniasis in South America [24–27]. *Endotrypanum* originally includes intra-erythrocytic trypanosomatids isolated from sloth *Choloepus didactylus* (Linnaeus) (Mesnil and Brimont 1908) and the transfer of *Le. colombiensis* to this genus is still debated, since their ability to infect erythrocytes has not been proven yet [28]. Furthermore, the role of sand flies as vector of these monoxenous and dixenous parasites to humans need to be investigated, although some sand flies known as permissive vectors seems able to sustain the infection with trypanosomatids under laboratory conditions [29–31].

Intense urbanization over the last decades has caused changes in natural habitats, especially those occupied by phlebotomine sand flies. Some species are able to withstand several environmental modifications that may occur in natural breeding sites and successfully adapt to urban environments [1]. In some regions, the presence of vector species in the human environment may be associated with the emergence of autochthonous cases of leishmaniases, as occurs in the state of Minas Gerais, where both VL and TL are widely spread [32,33]. In this context, the municipality of Brumadinho has reported an increasing number of VL and TL-cases over the last years, and some of them have been reported in the district of Casa Branca (Source: Municipal Secretary of Health of Brumadinho, MG). In this area, the presence of *Leishmania* spp. in wild and synanthropic mammals have been recorded [34]. Given this situation, this study aimed to describe the sand fly fauna, detect and identify by molecular assay the presence of trypanosomatids in females sand flies from Casa Branca.

## Material and methods

### Ethic statements

The collections were carried out with the authorization of the Sistema de Autorização e Informação em Biodiversidade—SISBIO (license number 15237–2). The owners of the residences verbally gave permission to the installation of the traps during the study. The sand flies were deposited in the Coleção de Flebotomíneos of the Instituto René Rachou (Fiocruz/COLFLEB).

**Study area, collection and identification of phlebotomine sand flies.** The study was conducted in the district of Casa Branca (20˚6'2.58"S; 44˚2'59.45" W), northern region of the municipality of Brumadinho (Fig 1). Casa Branca is bordered by the Parque Estadual Serra do Rola Moça (PESRM) conservation unit, the third largest urban park in Brazil, covering a transition area between Atlantic Forest and Cerrado. About eight kilometers west of Casa Branca is a village called Córrego do Feijão, where, in January 2019, an ore tailing dam ruptured, causing great environmental and social impact in all areas of Brumadinho municipality, as well as the death of 270 people.

Sand flies were collected from nine sampling points in Casa Branca from May 2013 to July 2014. Bimonthly, two automatic light traps (model HP) [35] were set at each sampling point,

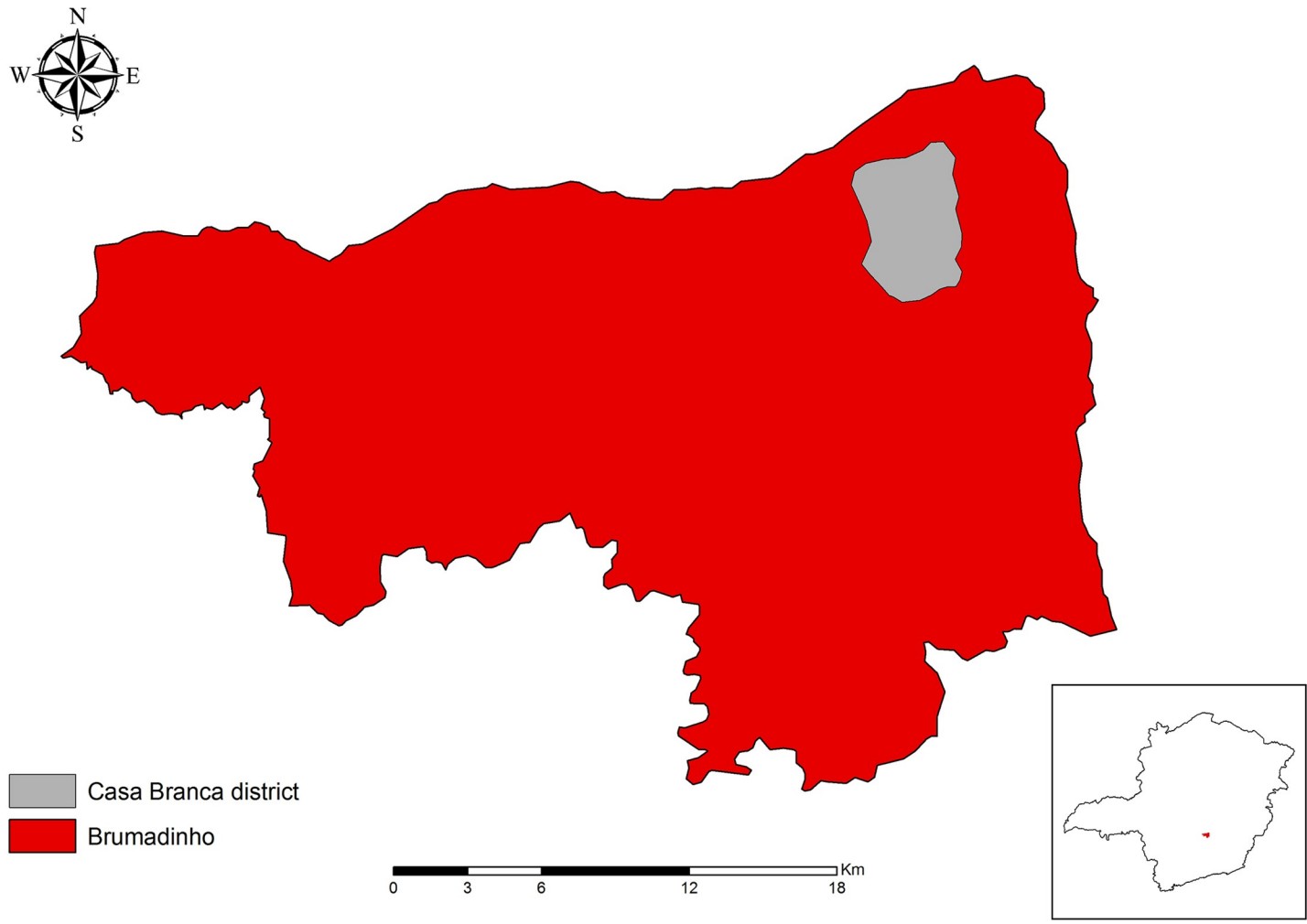

**Fig 1. Location of the study area in the district of Casa Branca, Brumadinho, Minas Gerais, Brazil.**

in the peridomiciliary environment for three consecutive nights. Collection sites with previous report of human or canine leishmaniasis were selected.

Collected sand flies were taken to the laboratory for further screening by sex, clarification process and dissection. All sand flies collected in July/2013, November/2013, March/2014 and July/2014 were mounted in Canada balsam on a glass slide for species identification. The female sand flies from May/2013, September/2013, January/2014 and May/2014 were dissected by removing the last three segments of the abdomen and head, mounted on a glass slide in Berlese liquid. The remainder of the body was individually stored dry at -20°C in a 1.5 mL tube for further DNA extraction.

Species identification was determined using the updated version of the classification proposed by Galati in 2003 [36]. The females of *Evandromyia sallesi* and *Evandromyia cortelezzii*, were identified as belonging to the "Cortelezzii complex" since they are undistinguishable by their morphology [37]. The abbreviation of the genera followed the proposal of Marcondes [38].

**DNA extraction and identification of trypanosomatids.** Females sand flies without blood in their abdomen were processed individually or grouped into pools of up to 20

individuals of the same species, date and site of collection. Females were subjected to DNA extraction using the Gentra Puregene® kit (Qiagen, Valencia, CA, USA) following the manufacturer's protocol. To control cross-contamination during DNA extraction, males sand flies were used as negative controls in all procedures [39,40]. In addition, all instruments and the entire work area were previously treated with DNAZap (Ambion, Life Technologies, Inc.).

The extracted DNA was used to investigate the presence of trypanosomatids by amplifying the internal transcribed spacer 1 region (ITS1), following the conditions described previously [41,42]. Positive controls were used in all PCR-reactions, as follows: *Leishmania amazonensis* (IFLA/BR/67/PH8), *Le. braziliensis* (MHOM/BR/75/M2903), *Le. infantum* (MHOM/BR/74/PP75), *Le. guyanensis* (MHOM/BR/75/M4147), *Crithidia fasciculata* (Fiocruz-COLPROT 048), *Endotrypanum monterogeii* (Fiocruz-COLPROT 151), *Herpetomonas samuelpessoai* (Fiocruz-COLPROT 067), *Leptomonas collosoma* (Fiocruz-COLPROT 073) and *Phytomonas serpens* (Fiocruz-COLPROT 189). The non-*Leishmania* trypanosomatids controls were kindly provided by the Protozoa Collection of the Laboratório de Biologia Molecular e Doenças Endêmicas (Fiocruz/COLPROT). The amplicons were visualized in a 2% agarose gel stained with ethidium bromide (10mg/mL) with a 100 bp DNA Step Ladder provided as molecular length size standard. The PCR-positive products were digested using *Hae*III enzyme following the conditions previously described [41]. The amplicons were purified with QIAquick® PCR Purification Kit (Qiagen, Valencia, CA, USA) as described by the manufacturer. Once purified, the products were used for DNA sequencing following the conditions described by Rêgo et al. [43]. Geneious software (v. 9.1.3) was used to check the electropherograms, align sequences and to perform comparison with sequences deposited in the GenBank database through BLAST alignment algorithm, considering the two most similar organisms according to identity above 90% for genus level and 98% for species level. These sequences are available in GenBank (Accession numbers MN638721-MN638737).

The positive rate of each sand fly species was determined considering only one specimen per positive pool.

**Statistical analyses.** To evaluate species abundance the index of species abundance "ISA", and the standardized index of species abundance "SISA" were calculated [44]. The values of SISA range from 0 to 1, with values closer to 1 representing the most abundant species. The Shannon index (H) and evenness index (J) [45] were used to determine species diversity and evenness of species abundance, respectively.

## Results

### Sand fly fauna

During the study period, 16,771 sand flies of 23 species and eight genera were collected in Casa Branca. The most abundant species was *Ny. whitmani* (70.9%; SISA 0.74), followed by *Lu. longipalpis* (15.2%; SISA 0.53) and *Mg. migonei* (9.1%; SISA 0.51) (Table 1 and Fig 2). The Shannon (H) and evenness (J) indexes were low for the study area (H' = 0.9946229; J' = 0.308997).

*Evandromyia* was the richest genera with eight species collected, followed by *Pintomyia* with six species (Table 1). May 2013 showed the greatest species abundance (7,054 sand flies) while July 2014 showed the lowest (266 sand flies). The highest species richness was observed in May 2014 with a total of 18 species (Table 1).

### Molecular identification of trypanosomatids

Of a total of 4,913 females, 47 were individually tested and 4,866 grouped in 311 pools. Table 2 presents information about the species collected and their distributions for molecular analysis. No sample individually analyzed was PCR positive for trypanosomatids.

**Table 1. Sand flies collected according to month at the Casa Branca, Brumadinho, Minas Gerais, from May 2013 to July 2014.**

| Species | Months of collection | | | | | | | | | | | | | | | | | |
|---|---|---|---|---|---|---|---|---|---|---|---|---|---|---|---|---|---|---|
| | May/13 | | Jul/14 | | Sep/13 | | Nov/13 | | Jan/14 | | Mar/14 | | May/14 | | Jul/14 | | Total | | |
| | ♂ | ♀ | ♂ | ♀ | ♂ | ♀ | ♂ | ♀ | ♂ | ♀ | ♂ | ♀ | ♂ | ♀ | ♂ | ♀ | ♂ | ♀ | Total (%) |
| *Brumptomyia* sp. | 0 | 0 | 0 | 1 | 4 | 4 | 0 | 0 | 0 | 0 | 0 | 2 | 0 | 0 | 0 | 0 | 4 | 7 | 11 (0.1) |
| Cortelezzii complex | 0 | 7 | 0 | 2 | 0 | 12 | 0 | 5 | 0 | 1 | 0 | 1 | 0 | 9 | 0 | 1 | 0 | 38 | 38 (0.2) |
| *Evandromyia cortelezzii* | 1 | 0 | 2 | 0 | 3 | 0 | 1 | 0 | 2 | 0 | 1 | 0 | 0 | 0 | 0 | 0 | 10 | 0 | 10 (0.1) |
| *Evandromyia edwarsi* | 0 | 0 | 0 | 0 | 0 | 0 | 0 | 0 | 0 | 0 | 0 | 0 | 0 | 1 | 0 | 0 | 0 | 1 | 1 (0.0) |
| *Evandromyia evandroi* | 0 | 0 | 0 | 0 | 0 | 0 | 0 | 0 | 1 | 0 | 0 | 0 | 0 | 0 | 0 | 0 | 1 | 0 | 1 (0.0) |
| *Evandromyia lenti* | 1 | 0 | 0 | 1 | 2 | 2 | 10 | 5 | 39 | 29 | 0 | 2 | 2 | 3 | 0 | 1 | 54 | 43 | 97 (0.6) |
| *Evandromyia sallesi* | 1 | 0 | 0 | 0 | 2 | 0 | 0 | 0 | 0 | 0 | 0 | 0 | 1 | 0 | 1 | 0 | 5 | 0 | 5 (0.0) |
| *Evandromyia teratodes* | 0 | 0 | 0 | 0 | 0 | 0 | 0 | 0 | 0 | 0 | 0 | 0 | 0 | 1 | 0 | 0 | 0 | 1 | 1 (0.0) |
| *Evandromyia termitophila* | 0 | 1 | 0 | 1 | 0 | 1 | 0 | 5 | 0 | 0 | 0 | 0 | 0 | 3 | 0 | 0 | 0 | 11 | 11 (0.1) |
| *Evandromyia tupynambai* | 0 | 1 | 0 | 0 | 3 | 0 | 0 | 2 | 0 | 2 | 0 | 0 | 1 | 10 | 0 | 0 | 4 | 15 | 19 (0.1) |
| *Lutzomyia amarali* | 0 | 0 | 0 | 0 | 0 | 0 | 0 | 0 | 0 | 0 | 0 | 2 | 0 | 1 | 0 | 0 | 0 | 3 | 3 (0.0) |
| *Lutzomyia ischyracantha* | 0 | 0 | 0 | 0 | 1 | 0 | 0 | 0 | 0 | 0 | 0 | 0 | 0 | 0 | 0 | 0 | 1 | 0 | 1 (0.0) |
| *Lutzomyia longipalpis* | 528 | 127 | 57 | 5 | 430 | 112 | 181 | 41 | 63 | 22 | 214 | 62 | 547 | 104 | 55 | 3 | 2,075 | 476 | 2,551 (15.2) |
| *Migonemyia migonei* | 219 | 90 | 72 | 12 | 175 | 53 | 156 | 49 | 104 | 17 | 73 | 24 | 373 | 62 | 41 | 13 | 1213 | 320 | 1533 (9.1) |
| *Nyssomyia intermedia* | 0 | 0 | 1 | 2 | 1 | 0 | 0 | 0 | 0 | 0 | 0 | 0 | 0 | 0 | 0 | 0 | 2 | 2 | 4 (0.0) |
| *Nyssomyia whitmani* | 3,449 | 2,535 | 79 | 263 | 793 | 680 | 547 | 343 | 625 | 324 | 101 | 72 | 974 | 959 | 77 | 62 | 6,645 | 5,238 | 11,883 (70.9) |
| *Pintomyia bianchigalatiae* | 0 | 13 | 0 | 0 | 0 | 10 | 1 | 2 | 0 | 3 | 0 | 17 | 0 | 30 | 0 | 0 | 1 | 75 | 76 (0.5) |
| *Pintomyia fischeri* | 7 | 13 | 6 | 2 | 7 | 11 | 34 | 15 | 27 | 34 | 14 | 18 | 11 | 49 | 5 | 4 | 111 | 146 | 257 (1.5) |
| *Pintomyia mamedei* | 0 | 1 | 0 | 0 | 0 | 0 | 0 | 0 | 0 | 0 | 0 | 0 | 0 | 0 | 0 | 0 | 0 | 1 | 1 (0.0) |
| *Pintomyia misionensis* | 0 | 0 | 0 | 3 | 0 | 0 | 0 | 0 | 0 | 1 | 0 | 0 | 0 | 4 | 0 | 1 | 0 | 9 | 9 (0.1) |
| *Pintomyia monticola* | 1 | 4 | 0 | 2 | 0 | 1 | 2 | 0 | 0 | 1 | 0 | 6 | 2 | 21 | 1 | 0 | 6 | 35 | 41 (0.2) |
| *Pintomyia pessoai* | 1 | 1 | 0 | 0 | 1 | 18 | 0 | 19 | 3 | 0 | 1 | 0 | 1 | 3 | 0 | 0 | 7 | 41 | 48 (0.3) |
| *Psathyromyia pascalei* | 0 | 0 | 0 | 0 | 1 | 0 | 0 | 0 | 0 | 1 | 0 | 0 | 1 | 0 | 1 | 0 | 3 | 1 | 4 (0.0) |
| *Psathyromyia* sp. | 0 | 0 | 0 | 1 | 0 | 0 | 0 | 0 | 0 | 0 | 0 | 0 | 0 | 0 | 0 | 0 | 0 | 1 | 1 (0.0) |
| *Psychodopygus lloydi* | 1 | 33 | 0 | 4 | 0 | 0 | 6 | 3 | 4 | 47 | 0 | 0 | 6 | 38 | 0 | 0 | 17 | 125 | 142 (0.9) |
| **Total** | 4,209 | 2,845 | 217 | 299 | 1,423 | 908 | 938 | 489 | 868 | 482 | 404 | 206 | 1,919 | 1,298 | 181 | 85 | 10,159 | 6,612 | 16,771 (100) |
| | **7,054** | | **516** | | **2,331** | | **1427** | | **1,350** | | **610** | | **3,217** | | **266** | | **16,771** | | |

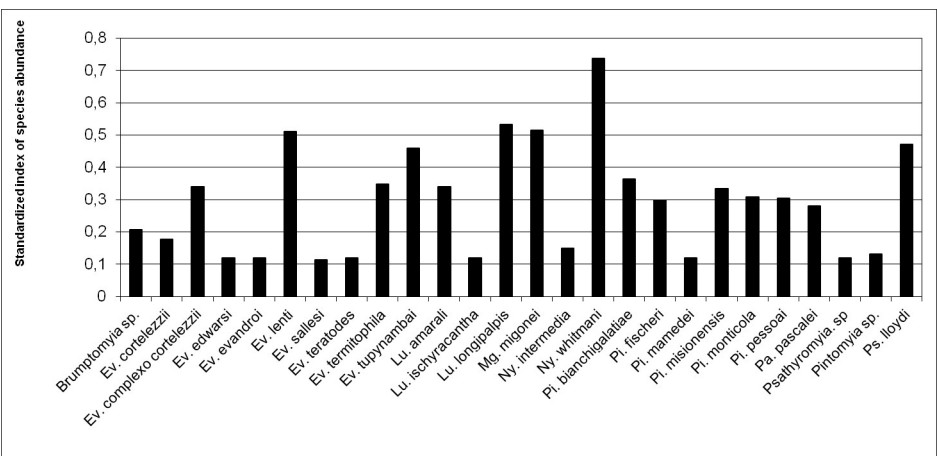

**Fig 2. Standardized index of species abundance of species collected at Casa Branca, Brumadinho, Minas Gerais from May 2013 to July 2014.**

**Table 2. Sand fly organized by species that were analyzed individually and in pools collected in Casa Branca, Brumadinho, Minas Gerais, from May 2013 to May 2014.**

| Species | Sand flies collected | Number of analyzed pools | Specimens analyzed individually |
|---|---|---|---|
| *Brumptomyia* sp* | 4 | 1 | 0 |
| Cortelezzii complex | 25 | 5 | 3 |
| *Evandromyia edwardsi* | 1 | 0 | 1 |
| *Evandromyia lenti* | 31 | 4 | 3 |
| *Evandromyia teratodes* | 1 | 0 | 1 |
| *Evandromyia termitophila* | 6 | 1 | 4 |
| *Evandromyia tupynambai* | 13 | 2 | 1 |
| *Lutzomyia amarali* | 1 | 0 | 1 |
| *Lutzomyia longipalpis* | 248 | 19 | 0 |
| *Migonemyia migonei* | 144 | 14 | 0 |
| *Nyssomyia whitmani* | 4128 | 220 | 0 |
| *Pintomyia bianchigalatiae* | 47 | 8 | 2 |
| *Pintomyia fischeri* | 80 | 13 | 0 |
| *Pintomyia misionensis* | 4 | 0 | 4 |
| *Pintomyia monticola* | 27 | 3 | 3 |
| *Pintomyia pessoai* | 16 | 3 | 1 |
| *Pintomyia* sp.* | 21 | 0 | 21 |
| *Psathyromyia pascalei* | 1 | 0 | 1 |
| *Psychodopygus lloydi* | 115 | 18 | 1 |
| **Total** | **4913** | **311** | **47** |

* sand flies damaged

Of the 311 pools tested, 13 (4.1%) were PCR-positive for *Leishmania*: *Lu. longipalpis* (1), *Ny. whitmani* (11) and *Ps. lloydi* (1) (Table 3). The positive rate obtained for *Leishmania* was 4.18% (13/311). Four pools of *Ny. whitmani* were positive for *Le. amazonensis* (4/311 = 1.2%), however, in other seven pools of this sand fly (7/311 = 2.2%) it was not possible to determinate the *Leishmania* species, due to low quality of the sequences, which contained a high number of ambiguous sites. Therefore, sequences were left as undetermined (*Leishmania* sp.). One pool of *Ps. lloydi* was positive for *Le. braziliensis* (1/311 = 0.3%) and *Lutzomyia longipalpis* was positive for *Leishmania* sp. (1/311 = 0.3%) (Table 3).

A total of five pools (5/311 = 1.6%) presented single amplicons of inconsistent size (400–500 bp), distinct to that predicted for *Leishmania* parasites (300–350 bp). These results led us to investigate the identity of these fragments and trypanosomatid species were identified. *Nyssomyia whitmani* and Cortelezzii complex (both one pool) were positive for *Herpetomonas* spp., while *Psychodopygus lloydi* (one pool) and *Ny. whitmani* (two pools) were positive for *Crithidia* spp. (Table 3). Sequence similarity to GenBank sequences ranged between 90–97%. The five non-*Leishmania* sequences found in our study, *Herpetomonas* spp. and *Crithidia* spp., are sufficiently different between them and possibly represent three and two, respectively, different MOTUs (Molecular Operational Taxonomic Units).

## Discussion

The sand fly diversity found in Casa Branca may be explained by the proximity to the PESRM conservation unit. The residences selected are also close to forest environments and presented breeding sites for phlebotomine sand flies, such as vegetation (fruit trees) and animal shelters (hennery, pigsty and dog kennel). The highest standardized index of species abundance (SISA)

**Table 3. Results of DNA detection and identification of species of Trypanosomatidae according to species of sand fly, number of pool and collection date in Casa Branca, Brumadinho from May 2013 to May 2014.**

| Sand fly species | Seq ID Blast | % similarity | Collection date | Accession numbers |
|---|---|---|---|---|
| Cortelezzii complex | *Herpetomonas* sp. | 95,38 | may-2013 | MN638722 |
| *Lutzomyia longipalpis* | *Leishmania* sp. | 80,2 | may-2013 | MN638733 |
| *Nyssomyia whitmani* | *Leishmania amazonensis* | 100 | may-2013 | MN638726 |
| *Nyssomyia whitmani* | *Leishmania amazonensis* | 99,39 | may-2013 | MN638738 |
| *Nyssomyia whitmani* | *Leishmania amazonensis* | 98,01 | sep-2013 | MN638734 |
| *Nyssomyia whitmani* | *Leishmania amazonensis* | 100 | may-2013 | MN638727 |
| *Nyssomyia whitmani* | *Leishmania* sp. | 99,51 | may-2013 | MN638724 |
| *Nyssomyia whitmani* | *Leishmania* sp. | 99,6 | may-2013 | MN638725 |
| *Nyssomyia whitmani* | *Leishmania* sp. | 99,66 | may-2013 | MN638731 |
| *Nyssomyia whitmani* | *Leishmania* sp. | 99,32 | may-2013 | MN638732 |
| *Nyssomyia whitmani* | *Leishmania* sp. | 99,04 | may-2013 | MN638735 |
| *Nyssomyia whitmani* | *Leishmania* sp. | 98,45 | may-2013 | MN638736 |
| *Nyssomyia whitmani* | *Leishmania* sp. | 98,72 | may-2013 | MN638737 |
| *Nyssomyia whitmani* | *Herpetomonas* sp. | 90,21 | may-2013 | MN638723 |
| *Nyssomyia whitmani* | *Herpetomonas* sp. | 97 | may-2013 | MN638729 |
| *Nyssomyia whitmani* | *Crithidia* sp. | 97,31 | may-2013 | MN638721 |
| *Psychodopygus lloydii* | *Leishmania braziliensis* | 99,32 | may-2014 | MN638730 |
| *Psychodopygus lloydii* | *Crithidia* sp. | 96,5 | may-2014 | MN638728 |

Seq ID Blast = hit provided by Blast search.

Trypanosomatid sequence obtained from strain A9 from the Protozoa Collection (Fiocruz/COLPROT)

in this study was found for *Ny. whitmani* (0.74). Despite the quite diversity of the study area (H ' = 0.995), the species abundances were not similar, in which *Ny. whitmani* was a dominant species. This likely explains the low value for the evenness index (J ' = 0.309), since the great abundance of just one species tends to impact the uniformity of the other ones.

Here, *Ny. whitmani* was found in high density close to residences. This species frequently has been found in endemic areas for TL in southeastern Brazil, and there is no doubt about their role as vector of *Le. braziliensis* [46]. Furthermore, *Ny. whitmani* has also been found harboring *Le. infantum* DNA in the state of Minas Gerais [47,48]. The presence *Le. amazonensis* DNA within *Ny. whitmani* has been previously reported in the states of Tocantins and Maranhão [49,50]. Changes in the natural environment may triggered a new epidemiological profile of leishmaniases that could involve *Leishmania amazonensis*, an alternative vector (i.e. *Ny. whitmani*) and new reservoirs (domestic and wild). Fonteles et al. [51] reports the capacity of *Ny. whitmani* to sustain infections with *Le. amazonensis*, suggesting their role as vector of this parasite in the state of Maranhão. In this context, as we shown here, the same may be occurring in Casa Branca where this sand fly was found harboring *Le. amazonensis* DNA.

*Lutzomyia longipalpis*, the main vector of *Le. infantum* in Brazil, was the second most abundant species in Casa Branca (15.22%) and was also found to be frequent close to the residences. This finding reinforces the adaptation of this species to human dwellings and peridomiciliary sites [52]. *Lu. longipalpis* is able to sustain infections with several *Leishmania* species [53–56] and their role as the main vector of *Le. infantum* in Latin America is undeniable [2,4]. The finding of *Leishmania* DNA within *Lu. longipalpis* from Casa Branca draw attention specially about cases of both human and canine visceral leishmaniasis reports.

*Migonemyia migonei* was the third most captured species in the study area with 9.14% and is also extremely anthropophilic species and frequently found inhabiting peridomiciliary sites

and feeding in domestic animals [57]. Recent studies have added information about the vector capacity of *Mg. migonei*. Guimarães et al. have reported their capacity to support *Le. infantum* development and late-stage infection [58] and moreover, natural infections by *Le. infantum* have been reported especially in the absence of the main vector, *Lu. longipalpis* [59–61]. *Migonemyia migonei* has also been implicated in the cycle of *Le. braziliensis* [62–64] and their capacity to support infections with this parasite have been confirmed by Alexandre et al [65].

The genus *Evandromyia* presented the highest number of species and although there are no proven vector species within this genus, natural infection or *Leishmania* DNA has been detected in some species [66,67]. In studies conducted in Minas Gerais, Carvalho et al. [68] detected *Le. infantum* DNA in *Ev. cortelezzii* and Saraiva et al. [69] found females of *Ev. sallesi* naturally infected by the same species of *Leishmania*. These findings may indicate that these species of the Cortelezzii complex (females of *Ev. cortelezzii* and *Ev. sallesi* do not differ morphologically) hold some epidemiological significance.

The species *Ps. lloydi* has been associated with the transmission of *Le. braziliensis* in wild areas of Minas Gerais [70,71] and was also found in this state with DNA of *Le. infantum* [72]. *Psychodopygus lloydi* was positive for DNA of *Leishmania braziliensis*, suggesting that this sand fly could maintain a wild cycle of TL in the District of Casa Branca.

The finding of *Herpetomonas* and *Crithidia* DNA within sand fly species from Casa Branca need to be investigated. Possibly, more than one species within each genera were detected, however, due to the small length of the amplicon it was not possible to confirm it. On the other hand, the molecular detections presented here, may represented different populations (different MOTUs), reinforcing the importance of further studies on these trypanosomatids. Although *Herpetomonas* seems not to be of epidemiological importance, it has been reported in Egyptian rat [73] and even in immunodepressed humans [18]. In their more common hosts, the Diptera, promastigotes live in the digestive tract, preferentially in the rectum, attached either to endothelian cells or as free-swimmers among fewer opisthomastigotes [74]. *Crithidia*-related parasites have been involved in an atypical manifestation similar to VL in Brazil [19]. The report of *Crithidia* causing clinical manifestations similar to VL and the detection of this parasites within sand flies from Brazil, as we found here and previously reported by Machado et al. [49] draw attention and need to be investigated.

Although in some positive pools were not possible to determinate the *Leishmania* species, due to the low quality of the sequences, it is noteworthy that there was no information about the sand fly fauna and putative vectors in Casa Branca. Here we provide consistent data that should be used for further investigations, such as trying to isolate *Le. amazonensis* from *Ny. whitmani* to confirm their role as a vector. Moreover, it is extremely important to investigate the presence of trypanosomatids in Casa Branca and their association with the sand fly fauna.

## Conclusion

This is the first study of a sand fly fauna in the municipality of Brumadinho, Minas Gerais. Our findings show that the Casa Branca locality has a diverse sand fly fauna with species that have been previously reported in the state of Minas Gerais. The *Ny. whitmani* species is the probable vector of *Le. braziliensis* in Casa Branca and may also be involved in the transmission of *Le. amazonensis*. The knowledge of the interactions between sand flies and trypanosomatids reported in this study shows that the infection may be occurring in the peridomiciliary environment in the study area. In addition, these results help to understand the dynamics of the leishmaniases transmission cycle in Casa Branca providing support for disease control actions in the region. The presence of non-*Leishmania* trypanosomatids raises an issue that has been neglected and is of great importance, the circulation of these parasites within phlebotomine sand flies.

## Acknowledgments

We thank Surveillance System in Health of Brumadinho for providing epidemiological data. We thank the residents of the collection sites for making the work possible.

## Author Contributions

**Conceptualization:** Aline Tanure, Felipe Dutra Rêgo, Célia Maria Ferreira Gontijo, Gustavo Fontes Paz, José Dilermando Andrade-Filho.

**Formal analysis:** Aline Tanure, Gabriel Barbosa Tonelli, Aldenise Martins Campos, Paloma Helena Fernandes Shimabukuro, Célia Maria Ferreira Gontijo, Gustavo Fontes Paz, José Dilermando Andrade-Filho.

**Funding acquisition:** José Dilermando Andrade-Filho.

**Investigation:** Aline Tanure, Felipe Dutra Rêgo, Gabriel Barbosa Tonelli, Aldenise Martins Campos, Paloma Helena Fernandes Shimabukuro, José Dilermando Andrade-Filho.

**Methodology:** Aline Tanure, Felipe Dutra Rêgo, Aldenise Martins Campos, Paloma Helena Fernandes Shimabukuro, Célia Maria Ferreira Gontijo, Gustavo Fontes Paz, José Dilermando Andrade-Filho.

**Writing – original draft:** Aline Tanure.

**Writing – review & editing:** Aline Tanure, Felipe Dutra Rêgo, Gabriel Barbosa Tonelli, Aldenise Martins Campos, Paloma Helena Fernandes Shimabukuro, Célia Maria Ferreira Gontijo, Gustavo Fontes Paz, José Dilermando Andrade-Filho.

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
