## [Editor Report · Decision Letter 0]

20 Mar 2020

PONE-D-20-05836

Diversity of phlebotomine sand flies and molecular detection of trypanosomatids species in Brumadinho, Minas Gerais, Brazil

PLOS ONE

Dear Dr. Andrade Filho,

Thank you for submitting your manuscript to PLOS ONE. After careful consideration, we feel that it has merit but does not fully meet PLOS ONE’s publication criteria as it currently stands. Therefore, we invite you to submit a revised version of the manuscript that addresses the points raised during the review process.

We would appreciate receiving your revised manuscript by May 04 2020 11:59PM. To enhance the reproducibility of your results, we recommend that if applicable you deposit your laboratory protocols in protocols.io, where a protocol can be assigned its own identifier (DOI) such that it can be cited independently in the future. For instructions see: http://journals.plos.org/plosone/s/submission-guidelines#loc-laboratory-protocols

We look forward to receiving your revised manuscript.

Kind regards,

Vyacheslav Yurchenko

Academic Editor

PLOS ONE

Additional Editor Comments (if provided):

Reading the manuscript of Tanure at al I noticed the huge unbalance in the literature citations. Authors (almost exclusively) cite Brazilian papers effectively ignoring the huge amount of work on this topic, which has been done in other parts of the world. I invite authors to correct this discrepancy and submit a revised version for peer review. V, Yurchenko

Journal Requirements:

2. One of the noted authors is a group or consortium [Grupo de Estudos em Leishmanioses]. In addition to naming the author group, please list the individual authors and affiliations within this group in the acknowledgments section of your manuscript. Please also indicate clearly a lead author for this group along with a contact email address.

4. Please ensure that you refer to Figure 2 in your text as, if accepted, production will need this reference to link the reader to the figure.

5. We note that Figure 1 in your submission contain map images which may be copyrighted. All PLOS content is published under the Creative Commons Attribution License (CC BY 4.0), which means that the manuscript, images, and Supporting Information files will be freely available online, and any third party is permitted to access, download, copy, distribute, and use these materials in any way, even commercially, with proper attribution. For these reasons, we cannot publish previously copyrighted maps or satellite images created using proprietary data, such as Google software (Google Maps, Street View, and Earth). For more information, see our copyright guidelines: http://journals.plos.org/plosone/s/licenses-and-copyright.
---

## [Author Response · Author response to Decision Letter 0]

22 Apr 2020

PONE-D-20-05836

Diversity of phlebotomine sand flies and molecular detection of trypanosomatids species in Brumadinho, Minas Gerais, Brazil

PLOS ONE

Dear Vyacheslav Yurchenko,

Below are our answers to requests from the Editor and the PLOS ONE technical team.

Thanks

Additional Editor Comments (if provided):

Reading the manuscript of Tanure at al I noticed the huge unbalance in the literature citations. Authors (almost exclusively) cite Brazilian papers effectively ignoring the huge amount of work on this topic, which has been done in other parts of the world. I invite authors to correct this discrepancy and submit a revised version for peer review. V, Yurchenko

The Editor is right, there were many references in Portuguese that could be replaced by others, thanks for the suggestion.

Journal Requirements:

We understand that our article and files are based on the PLOS ONE style.

2. One of the noted authors is a group or consortium [Grupo de Estudos em Leishmanioses]. In addition to naming the author group, please list the individual authors and affiliations within this group in the acknowledgments section of your manuscript. Please also indicate clearly a lead author for this group along with a contact email address.

Grupo de Estudos em Leishmanioses is the author's affiliation, a name similar to "laboratory". All authors belong to the leishmaniasis study group (Grupo de Estudos em Leishmanioses)

 All authors are listed and with their affiliation

4. Please ensure that you refer to Figure 2 in your text as, if accepted, production will need this reference to link the reader to the figure.

 We have included the reference to figure 2 in the text, line 214

5. We note that Figure 1 in your submission contain map images which may be copyrighted. All PLOS content is published under the Creative Commons Attribution License (CC BY 4.0), which means that the manuscript, images, and Supporting Information files will be freely available online, and any third party is permitted to access, download, copy, distribute, and use these materials in any way, even commercially, with proper attribution. For these reasons, we cannot publish previously copyrighted maps or satellite images created using proprietary data, such as Google software (Google Maps, Street View, and Earth). For more information, see our copyright guidelines: http://journals.plos.org/plosone/s/licenses-and-copyright.

 We will provide copyright permission for figure 1 and include it in the submission system

Thank you!!!!

José Dilermando Andrade Filho

---

## [Decision Letter · Decision Letter 1]

4 May 2020

PONE-D-20-05836R1

Diversity of phlebotomine sand flies and molecular detection of trypanosomatids species in Brumadinho, Minas Gerais, Brazil

PLOS ONE

Dear Dr. Andrade Filho,

Thank you for submitting your manuscript to PLOS ONE. After careful consideration, we feel that it has merit but does not fully meet PLOS ONE’s publication criteria as it currently stands. Therefore, we invite you to submit a revised version of the manuscript that addresses the points raised during the review process.

We would appreciate receiving your revised manuscript by Jun 18 2020 11:59PM. To enhance the reproducibility of your results, we recommend that if applicable you deposit your laboratory protocols in protocols.io, where a protocol can be assigned its own identifier (DOI) such that it can be cited independently in the future. For instructions see: http://journals.plos.org/plosone/s/submission-guidelines#loc-laboratory-protocols

We look forward to receiving your revised manuscript.

Kind regards,

Vyacheslav Yurchenko

Academic Editor

PLOS ONE

Additional Editor Comments (if provided):

Please address all the concerns raised by 3 reviewers. In addition, I request: 1) to extend the introduction on trypanosomatids and discuss the following publications: Maslov et al, 2019 (Parasitology) and Lukes et al, 2018 (Trends Prasitol). 2) to spell all species names in Italic; iii) fix the references; iv) if you cite ref 60, please cite an accompanying paper- Kostygov et al, 2019 (Trop Med Int Health).

Reviewers' comments:

Reviewer's Responses to Questions

**Comments to the Author**

1. If the authors have adequately addressed your comments raised in a previous round of review and you feel that this manuscript is now acceptable for publication, you may indicate that here to bypass the “Comments to the Author” section, enter your conflict of interest statement in the “Confidential to Editor” section, and submit your "Accept" recommendation.

Reviewer #1: (No Response)

Reviewer #2: (No Response)

Reviewer #3: (No Response)

2. Is the manuscript technically sound, and do the data support the conclusions?

Reviewer #1: Partly

Reviewer #2: Yes

Reviewer #3: Yes

3. Has the statistical analysis been performed appropriately and rigorously? 

Reviewer #1: No

Reviewer #2: I Don't Know

Reviewer #3: Yes

4. Have the authors made all data underlying the findings in their manuscript fully available?

Reviewer #1: Yes

Reviewer #2: Yes

Reviewer #3: Yes

5. Is the manuscript presented in an intelligible fashion and written in standard English?

Reviewer #1: Yes

Reviewer #2: Yes

Reviewer #3: No

6. Review Comments to the Author

Reviewer #1: This is a nice study of a sand fly fauna (in the Casa Branca district of Brumadinho municipality) demonstrating the high phlebotomine diversity and the presence of DNA of two Leishmania species as well as several (3-4) non-Leishmania (monoxenous) parasites. However, because the design of the study (bimonthly = visiting of the locality once per two months) is not sufficient for any “ecological” analysis, I would suggest deleting all parts connected with “seasonal” variation in sand fly abundance and climate (and other) data. The number of collections is really not sufficient for any similar analysis; but even without these unreliable and therefore misleading results, the study is fully reliable to be published. However, there are several less important unclarities to must be solved.

4 trypanosomatid species or trypanosomatids

32 HP – meaning

33 better trap/nights (to demonstrate the effort)

39-40 how many individuals and how many in pools (BTW, pools of 1 specimen is not a pool !!!)

(Of a total of 4,913 females, 47 were individually tested and 4,866 grouped in 311 pools).

42 instead of Leishmania sp. use … in seven pools of Ny. whitmani and one pool of Lu. longipalpis the leishmania species was not determined (to the low quality of the sequences) … or something like this, otherwise, it seems that some “new/unknown” leishmania species was detected

43 .

44-45 same type of result presentation as in the case of Leishmania spp. – it means: how many pools/individuals and in which sand fly species.

45 The presence of DNA is not proving of vector capacity! (… may be involved in the transmission of Le. amazonensis … etc. is much more accurate)

55-57 REFs 1-3 are more-or-less relevant, but there are several current reviews, e.g., Akhoundi et al. 2016 (PLoS Neglected Tropical Diseases 10(3): e0004349.), etc… that would fit here.

59 REF 4 – ok, but add also e.g., Rangel, Shaw (2018) Brazilian Sand Flies … (10.1007/978-3-319-75544-1) etc.

71-72 The genus Trypanosoma is missing! Sand flies are possible vectors of several trypanosome species – add a relevant reference(s)

(the sentence “Several authors have reported the presence of non-Leishmania trypanosomatids DNA in sand flies by means of molecular techniques [12, 13, 14, 15]” does not cover this issue clear enough)

73 REFs 10 and 11 – OK, but add some references on Leptomonas (e.g. Kraeva et al. 2015 (PLoS Pathog 11(8):e1005127)); and even more references would be appropriate here

71-73 differentiate between monoxenous and dixenous genera

77-81 The statement is controversial in some aspects – first of all, the situation with the genus “Endotrypanum” is rather complicated (see doi: 10.14411/fp.2017.020). And in addition to this, the sentence, which is articulated in this way and is given in the previous contexts, makes an impression on readers that the authors refer primarily to single-host (monoxenous) genera (such as Crithidia, Leptomonas, etc.).

124 CDC light traps (model HP) – I am sure that there is no official model HP within the CDC traps – it is an only modification based on the CDC trap design – which company produces these traps or is it just some home-made version?

125-6 not clear – it means that the traps run three nights (and two days)?

Sand flies were still alive (after three days???), How many sand flies survive till laboratory (approximately in % ???)

137-8 when the morphological identification was made? The key is from the end of 2018, the collection and slide preparation was made much earlier (as was mentioned in the previous response to reviewers)!!!

158 … of trypanosomatids (not Leishmania)

247 – the prevalence is right, but anyway it should be explained that the authors predict that only one specimen per pool was infected… (it corresponds with the low “pool” prevalence)

253/255 – why “it was not possible get to species level”? - due to the low quality of the sequences ??? or the sequences represent some new leishmania species ??? It must be explained!

258-9 “the presence of bands of inconsistent size (400-500 bp) with fragments for Leishmania (~350 bp)” – not clear. Does it mean that the double bands were detected = mixed infection of leishmania (350bp) + some monoxenous trypanosomatids ??? And if yes, it means that monoxenous trypanosomatids occur only in pools positive for leishmania parasites???? It is really very unusual and must be discussed!

260 - … three trypanosomatids … - must be clearly distinguished even in Tab. 3 – e.g., Herpetomonas sp. 1, Herpetomonas sp. 2 …

263/352 – Herpetomonas sp. means unidentified ONE species of the genus Herpetomonas, but it seems that two different Herpetomonas species were detected – thus Herpetomonas spp.!!!! Or better cortezzii – H. sp. 1, Ny.w. – H. sp. 2, to make it clear. The same for Crithidia – it is not clear if Crithidia parasites from Ps.l. and Ny.w. belong to one or two species – if the sequences represent two species, it would be better to use C. sp. 1 and C. sp. 2 to make it clear. You have to make it clear, especially if you are not present any phylogenetic tree etc.

Table 3 – sort it according to “Host” or according to “trypanosomatid parasites!, not according Acc.Nos., it makes no sense.

Seq ID Blast – the different unnamed Herpetomonas/Crithidia/Leishmania species (identified by BLAST) must be specified (by its voucher name/number etc., or even by their GB Acc.No.)!

Reviewer #2: I appreciate that authors improved the manuscript and add more international references. However, some of them were not chosen correctly. I suggest correct several references in the first half of the Introduction:

Line 55: Ref. 2 (Alkan et al) is not an optimal reference. Would be better to include the famous review by Maroli et al. 2013, see the pdf.

Line 57: Ref. 3 (Lestinova et al) is not suitable reference. Again, the review by Michele Maroli could be used.

Line 59: Ref. 4. There is more recent review on this topic: chapter by Dvorak et al, published in Bruschi Fabrizio, Gradoni Luigi (eds.), The Leishmaniases: Old Neglected Tropical Diseases. Springer 2018.

Lines 67-70: Please use some more recent reference. All this information is described in details by Jeffrey Shaw in the review by Dvorak et al.

In Discussion, I suggest to improve discussion about Mg. migonei vector competence to L. infantum (Lines 292-294). The vectorial status of Mg. migonei was recently proven experimentally by Fitipaldi Veloso Guimaraes et al, 2016. Please, add this reference (or replace the existing one, no. 45).

Finally, I suggest shorten the “Short title” as it is too long now.

For more details and some other minor corrections of typos, see the pdf file with my comments.

Reviewer #3: The authors shall be commended for the extremely high number of determined and examined insects, and the solid way they approached this task. Massive undertaking, the results of which are worth publishing, but this should happen in somewhat different form and format.

I gather that it is difficult to write a readable paper out of huge but relatively boring data, but the authors still shall try to do a better job. In general, the authors describe frequently and extensively things that can be found on Wikipedia (geography, how many people live in some district - the number is non-scientific because at the time of writing it was already incorrect), please remove all this, which unncessarily extends the manuscript and brings absolutely no novelty and has no relevance to the presented data (where is a correlation between the size of the district and phlebotomine sandflies?).

I would like to ask the authors to better balance the Results and the Discussion, the first being too short and the latter too long.

Discussing usefulness or not of PCR-RFLP and restriction enzymes is way outdated, this type of discussions were concluded 15 or rather 20 years ago and is irrelevant nowadays, when it is sequencing or nothing. At least as long as you want to publish in respected journals, such as PloS One. Hence, shorten this part of discussion by 90%. Again, sentences such as “DNA sequencing has generated good results when used for detection of 380 Leishmania in sand flies“ are funny and make no sense these days, sequencing when done properly always (!) generates good results, namely the letter AGCT.

Furthermore, the acknowledgements and technical details are extremely long and tedious, nobody says that gels were stained with EtBr anymore, this is a common knowledge - please avoid this South American stretching of the manuscripts, remove what is the capital etc. etc.

Give more attention to the really interesting stuff regarding the parasites found. There is tons of references, which are overlaping with their info (+ IBGE reference – accessed in 2016? What is this?) This and some others are not standard type references and has to be deleted. Refs are in different fonts, formats, excessively long list describing all the time the same thing. Yet there is absolutely no discussion and literature about the potentially interesting findings of non-Leishmania trypanosomatids, which is a mistake that has to be rectified.

7. PLOS authors have the option to publish the peer review history of their article (what does this mean?). If published, this will include your full peer review and any attached files.

Reviewer #1: No

Reviewer #2: No

Reviewer #3: No

---

## [Author Response · Author response to Decision Letter 1]

15 May 2020

Manuscript ID: PONE-D-20-05836R1

Dear Dr. Yurchenko,

We would like to thank for your thoughtful review of the manuscript. You raised important issues and your inputs were well-considered and gratefully implemented. We agree with almost all your comments and we have revised our manuscript accordingly.

We have crafted a revised version of the paper stating the hypothesis and the implications of our work more clearly than before, including the suggestions. We are confident that the new version of the manuscript is greatly improved. We respond below in details to each of your comments. If we have slightly disagreed with something, we stated why. We hope that you will find our responses to your comments satisfactory, and we are willing to finish the revised version of the manuscript.

Please, find below your comments repeated in italics and our responses inserted after each comment. To facilitate the revision process, in some instances we refer to the plain text file indicating the page and the line (page-line). 

Looking forward to hearing from you soon.

Sincerely,

José Dilermando Andrade Filho

Additional Editor Comments (if provided): Please address all the concerns raised by 3 reviewers. In addition, I request: 1) to extend the introduction on trypanosomatids and discuss the following publications: Maslov et al, 2019 (Parasitology) and Lukes et al, 2018 (Trends Prasitol). 2) to spell all species names in Italic; iii) fix the references; iv) if you cite ref 60, please cite an accompanying paper- Kostygov et al, 2019 (Trop Med Int Health).

Authors’ comment: We accept all your suggestions. The introduction has been improved with the mentioned papers; all species names and the references have been revised as well. Information about the paper Kostygov et al. 2019 was also added in the manuscript (now lines 58-59).

Reviewer #1: This is a nice study of a sand fly fauna (in the Casa Branca district of Brumadinho municipality) demonstrating the high phlebotomine diversity and the presence of DNA of two Leishmania species as well as several (3-4) non-Leishmania (monoxenous) parasites. However, because the design of the study (bimonthly = visiting of the locality once per two months) is not sufficient for any “ecological” analysis, I would suggest deleting all parts connected with “seasonal” variation in sand fly abundance and climate (and other) data. The number of collections is really not sufficient for any similar analysis; but even without these unreliable and therefore misleading results, the study is fully reliable to be published. However, there are several less important unclarities to must be solved.

Authors’ comment: We totally agree with your comment. All data about the relationship between seasonal variation and sand fly abundance have been removed.

4 trypanosomatid species or trypanosomatids 

Authors’ comment: The title has been changed (now line 3).

32 HP – meaning

Authors’ comment: The sentence has been changed (now line 20-22).

33 better trap/nights (to demonstrate the effort)

Authors’ comment: The sentence has been changed (now lines 20-22).

39-40 how many individuals and how many in pools (BTW, pools of 1 specimen is not a pool !!!) (Of a total of 4,913 females, 47 were individually tested and 4,866 grouped in 311 pools).

Authors’ comment: The sentence has been changed (now lines 20-24).

42 instead of Leishmania sp. use … in seven pools of Ny. whitmani and one pool of Lu. longipalpis the leishmania species was not determined (to the low quality of the sequences) … or something like this, otherwise, it seems that some “new/unknown” leishmania species was detected 43.

Authors’ comment: The sentence has been changed (now lines 28-30).

44-45 same type of result presentation as in the case of Leishmania spp. – it means: how many pools/individuals and in which sand fly species.

Authors’ comment: The sentence has been changed (now lines 27-33).

45 The presence of DNA is not proving of vector capacity! (… may be involved in the transmission of Le. amazonensis … etc. is much more accurate)

Authors’ comment: The sentence has been changed (now line 34-36).

55-57 REFs 1-3 are more-or-less relevant, but there are several current reviews, e.g., Akhoundi et al. 2016 (PLoS Neglected Tropical Diseases 10(3): e0004349.), etc… that would fit here.

Authors’ comment: The references have been changed (now lines 43-46).

59 REF 4 – ok, but add also e.g., Rangel, Shaw (2018) Brazilian Sand Flies … (10.1007/978-3-319-75544-1) etc.

Authors’ comment: The reference has been added (now line 49).

71-72 The genus Trypanosoma is missing! Sand flies are possible vectors of several trypanosome species – add a relevant reference(s) (the sentence “Several authors have reported the presence of non-Leishmania trypanosomatids DNA in sand flies by means of molecular techniques [12, 13, 14, 15]” does not cover this issue clear enough)

Authors’ comment: The sentence has been changed and references covering this issue have been added (now lines 50-70).

73 REFs 10 and 11 – OK, but add some references on Leptomonas (e.g. Kraeva et al. 2015 (PLoS Pathog 11(8):e1005127)); and even more references would be appropriate here

Authors’ comment: The reference was added (now line 70).

71-73 differentiate between monoxenous and dixenous genera

Authors’ comment: This information has been added (now lines 50-53).

77-81 The statement is controversial in some aspects – first of all, the situation with the genus “Endotrypanum” is rather complicated (see doi: 10.14411/fp.2017.020). And in addition to this, the sentence, which is articulated in this way and is given in the previous contexts, makes an impression on readers that the authors refer primarily to single-host (monoxenous) genera (such as Crithidia, Leptomonas, etc.).

Authors’ comment: The paragraph has been deeply revised. Now, we believe that the background context that we provided make clear our goal (now lines 50-70).

124 CDC light traps (model HP) – I am sure that there is no official model HP within the CDC traps – it is an only modification based on the CDC trap design – which company produces these traps or is it just some home-made version?

Authors’ comment: We agree with your comment. The automatic light trap (model HP) is an improved model of suction light trap for the capture of small insects, similar to CDC light traps. However, now in the manuscript text, we referred to those traps as “automatic light traps (model HP)” and not CDC light traps (model HP). This information could be check in the lines 99-101.

125-6 not clear – it means that the traps run three nights (and two days)?

Sand flies were still alive (after three days???), How many sand flies survive till laboratory (approximately in % ???)

Authors’ comment: The sentence has been improved (now lines 99-102). Many sand flies still alive after three days, but we could not provide the percentage since it was not evaluated. Furthermore, to carry out molecular techniques, living sand flies is unnecessary.

137-8 when the morphological identification was made? The key is from the end of 2018, the collection and slide preparation was made much earlier (as was mentioned in the previous response to reviewers)!!!

Authors’ comment: The reference Galati (2018) is a book chapter including updates of Galati (2003). In the previous review, the changing of the oldest reference for a newest one was suggested, especially because it is a Portuguese version, so we did it. Even so, we rewrite the sentence highlighting this information (now lines 110-111).

158 … of trypanosomatids (not Leishmania)

Authors’ comment: The sentence was been changed (now line 116).

247 – the prevalence is right, but anyway it should be explained that the authors predict that only one specimen per pool was infected… (it corresponds with the low “pool” prevalence)

Authors’ comment: We totally agree with your comment. However, in the new version we presented the prevalence per positive pool (now lines 183-198).

253/255 – why “it was not possible get to species level”? - due to the low quality of the sequences ??? or the sequences represent some new leishmania species ??? It must be explained!

Authors’ comment: All sequences without species level identification presented low quality, containing high number of ambiguous sites. We stated this information in the lines 185-188.

258-9 “the presence of bands of inconsistent size (400-500 bp) with fragments for Leishmania (~350 bp)” – not clear. Does it mean that the double bands were detected = mixed infection of leishmania (350bp) + some monoxenous trypanosomatids ??? And if yes, it means that monoxenous trypanosomatids occur only in pools positive for leishmania parasites???? It is really very unusual and must be discussed!

Authors’ comment: We observed that the non-Leishmania parasites shown amplicons of 400-500 pb, distinct to predicted for Leishmania parasites (300-350pb). And this fact, led us to investigate what was amplified. There was no double-bands/mixed infections. However, the sentence has been changed (192-195).

260 - … three trypanosomatids … - must be clearly distinguished even in Tab. 3 – e.g., Herpetomonas sp. 1, Herpetomonas sp. 2 …

Authors’ comment: The sentence has been changed (now lines 193-195).

263/352 – Herpetomonas sp. means unidentified ONE species of the genus Herpetomonas, but it seems that two different Herpetomonas species were detected – thus Herpetomonas spp.!!!! Or better cortezzii – H. sp. 1, Ny.w. – H. sp. 2, to make it clear. The same for Crithidia – it is not clear if Crithidia parasites from Ps.l. and Ny.w. belong to one or two species – if the sequences represent two species, it would be better to use C. sp. 1 and C. sp. 2 to make it clear. You have to make it clear, especially if you are not present any phylogenetic tree etc.

Authors’ comment: The sentence was been changed (now lines 192-198). As the sequences have aligned with “Herpetomonas sp.” and “Crithidia sp.” on Blast tool, we are not sure about the presence of more than one species within both genera of trypanosomatids. So, we decide to keep “sp.” referring at least one species of each genera.

Table 3 – sort it according to “Host” or according to “trypanosomatid parasites!, not according Acc.Nos., it makes no sense.

Authors’ comment: We accepted your suggestion. The table 3 has been changed according hosts.

Seq ID Blast – the different unnamed Herpetomonas/Crithidia/Leishmania species (identified by BLAST) must be specified (by its voucher name/number etc., or even by their GB Acc.No.)!

Authors’ comment: We are not sure about your inquiry, since all species identified by Blast tool have already been identified by their accession numbers, as presented in Table 3.

Reviewer #2: I appreciate that authors improved the manuscript and add more international references. However, some of them were not chosen correctly. I suggest correct several references in the first half of the Introduction:

Authors’ comment: We accept your suggestion. We add more international references when it was possible, especially in the first half of the introduction. However, for several aspects about sand fly fauna is desirable to use national references.

Line 55: Ref. 2 (Alkan et al) is not an optimal reference. Would be better to include the famous review by Maroli et al. 2013, see the pdf.

Authors’ comment: The reference has been added (now line 43-46).

Line 57: Ref. 3 (Lestinova et al) is not suitable reference. Again, the review by Michele Maroli could be used.

Authors’ comment: The sentence has been changed (now line 46).

Line 59: Ref. 4. There is more recent review on this topic: chapter by Dvorak et al, published in Bruschi Fabrizio, Gradoni Luigi (eds.), The Leishmaniases: Old Neglected Tropical Diseases. Springer 2018.

Authors’ comment: The reference has been added (now line 49).

Lines 67-70: Please use some more recent reference. All this information is described in details by Jeffrey Shaw in the review by Dvorak et al.

Authors’ comment: The reference has been added (now line 49).

In Discussion, I suggest to improve discussion about Mg. migonei vector competence to L. infantum (Lines 292-294). The vectorial status of Mg. migonei was recently proven experimentally by Fitipaldi Veloso Guimaraes et al, 2016. Please, add this reference (or replace the existing one, no. 45).

Authors’ comment: Discussion about Mg. migonei has been improved (now lines 230-238).

Finally, I suggest shorten the “Short title” as it is too long now.

Authors’ comment: The short title has been reviewed (now line 1).

For more details and some other minor corrections of typos, see the pdf file with my comments.

Authors’ comment: All suggestions marked in pdf file have been accepted.

Reviewer #3: I gather that it is difficult to write a readable paper out of huge but relatively boring data, but the authors still shall try to do a better job. In general, the authors describe frequently and extensively things that can be found on Wikipedia (geography, how many people live in some district - the number is non-scientific because at the time of writing it was already incorrect), please remove all this, which unncessarily extends the manuscript and brings absolutely no novelty and has no relevance to the presented data (where is a correlation between the size of the district and phlebotomine sandflies?).

Authors’ comment: We accepted your suggestion. We did a deeply revision in the manuscript file.

I would like to ask the authors to better balance the Results and the Discussion, the first being too short and the latter too long.

Authors’ comment: We accepted your suggestion. We did a deeply revision in the manuscript file.

Discussing usefulness or not of PCR-RFLP and restriction enzymes is way outdated, this type of discussions were concluded 15 or rather 20 years ago and is irrelevant nowadays, when it is sequencing or nothing. At least as long as you want to publish in respected journals, such as PloS One. Hence, shorten this part of discussion by 90%. Again, sentences such as “DNA sequencing has generated good results when used for detection of 380 Leishmania in sand flies“ are funny and make no sense these days, sequencing when done properly always (!) generates good results, namely the letter AGCT.

Authors’ comment: We accepted your suggestion. We did a deeply revision in the discussion section.

Furthermore, the acknowledgements and technical details are extremely long and tedious, nobody says that gels were stained with EtBr anymore, this is a common knowledge - please avoid this South American stretching of the manuscripts, remove what is the capital etc. etc.

Authors’ comment: We totally disagree with your comment. With all due respect, we do not understand how 2-3 lines of acknowledgments are quite boring. This is a personal section and we thank everyone who we would like. About gel electrophoresis, the most used staining for sure is EtBr however, it is not the only one. Alternative DNA dyes, such as SYBR Safe is also currently used. So, we decide to keep this information because it is not trivial, and it is not a "South American stretching" as you said. However, as we said in the last inquiries, we carefully revised the manuscript and we hope this version stated better our goal.

Give more attention to the really interesting stuff regarding the parasites found. There is tons of references, which are overlaping with their info (+ IBGE reference – accessed in 2016? What is this?) This and some others are not standard type references and has to be deleted. Refs are in different fonts, formats, excessively long list describing all the time the same thing. Yet there is absolutely no discussion and literature about the potentially interesting findings of non-Leishmania trypanosomatids, which is a mistake that has to be rectified.

Authors’ comment: The discussion section has been improved. Now, we discuss about the finding of non-Leishmania trypanosomatids (now lines 252-261).

---

## [Decision Letter · Decision Letter 2]

19 May 2020

PONE-D-20-05836R2

Diversity of phlebotomine sand flies and molecular detection of trypanosomatids in Brumadinho, Minas Gerais, Brazil

PLOS ONE

Dear Dr. Andrade Filho,

Thank you for submitting your manuscript to PLOS ONE. After careful consideration, we feel that it has merit but does not fully meet PLOS ONE’s publication criteria as it currently stands. Therefore, we invite you to submit a revised version of the manuscript that addresses the points raised during the review process.

We would appreciate receiving your revised manuscript by Jul 03 2020 11:59PM. To enhance the reproducibility of your results, we recommend that if applicable you deposit your laboratory protocols in protocols.io, where a protocol can be assigned its own identifier (DOI) such that it can be cited independently in the future. For instructions see: http://journals.plos.org/plosone/s/submission-guidelines#loc-laboratory-protocols

We look forward to receiving your revised manuscript.

Kind regards,

Vyacheslav Yurchenko

Academic Editor

PLOS ONE

Additional Editor Comments (if provided):

Please address the final comments.

Reviewers' comments:

Reviewer's Responses to Questions

**Comments to the Author**

1. If the authors have adequately addressed your comments raised in a previous round of review and you feel that this manuscript is now acceptable for publication, you may indicate that here to bypass the “Comments to the Author” section, enter your conflict of interest statement in the “Confidential to Editor” section, and submit your "Accept" recommendation.

Reviewer #1: All comments have been addressed

Reviewer #3: All comments have been addressed

2. Is the manuscript technically sound, and do the data support the conclusions?

Reviewer #1: Yes

Reviewer #3: Yes

3. Has the statistical analysis been performed appropriately and rigorously? 

Reviewer #1: Yes

Reviewer #3: N/A

4. Have the authors made all data underlying the findings in their manuscript fully available?

Reviewer #1: Yes

Reviewer #3: Yes

5. Is the manuscript presented in an intelligible fashion and written in standard English?

Reviewer #1: Yes

Reviewer #3: Yes

6. Review Comments to the Author

Reviewer #1: I am more-or less satisfied with the current version, but I do not fully agree with one specific explanation.

Based on the sequences, which are already available in the GenBank database, it is clear that the sequences of monoxenous trypanosomatids differ from each other (within the three Herpetomonas spp. as well as between the two Crithidia spp.). Therefore, this information should be mentioned somewhere in the text, i.e., that the found Herpetomonas spp. and Crithidia spp. probably represent three and two, respectively, species, or at least different populations (or the Molecular Operational Taxonomic Unit; MOTU, could be used in this case), to make it clear that these sequences do not represent the same herpetomonas or Crithidia species. And, accordingly, Herpetomonas spp. and Crithidia spp. must be used in the text.

I would also recommend changing the sentence in the abstract

Instead of

“In seven pools of Ny. whitmani and in one pool of Lu. longipalpis the Leishmania species was not determined due to the low quality of the sequences.”

change like:

“In seven pools of Ny. whitmani and in one pool of Lu. longipalpis positive for leishmania DNA, the parasite species was not determined due to the low quality of the sequences.”

Reviewer #3: I find the way the authors responded acceptable and consider the paper much improved. I would still have some issues with the length of acknowledgements (theoretically you can have it as long as the discussion, there is no official limit, but it is simply not good), the way the gel is stained is irrelevant - it was stained somehow and that's all that matters, but I can live with the present version.

7. PLOS authors have the option to publish the peer review history of their article (what does this mean?). If published, this will include your full peer review and any attached files.

Reviewer #1: No

Reviewer #3: No

---

## [Author Response · Author response to Decision Letter 2]

20 May 2020

Manuscript ID: PONE-D-20-05836R1

Dear Dr. Yurchenko,

Dear Editor,

below is our response to the reviewers' comments.

We are available for clarification.

Thank you

Sincerely,

José Dilermando Andrade Filho

Reviewer #1: I am more-or less satisfied with the current version, but I do not fully agree with one specific explanation.

Based on the sequences, which are already available in the GenBank database, it is clear that the sequences of monoxenous trypanosomatids differ from each other (within the three Herpetomonas spp. as well as between the two Crithidia spp.). Therefore, this information should be mentioned somewhere in the text, i.e., that the found Herpetomonas spp. and Crithidia spp. probably represent three and two, respectively, species, or at least different populations (or the Molecular Operational Taxonomic Unit; MOTU, could be used in this case), to make it clear that these sequences do not represent the same herpetomonas or Crithidia species. And, accordingly, Herpetomonas spp. and Crithidia spp. must be used in the text.

I would also recommend changing the sentence in the abstract

Instead of

“In seven pools of Ny. whitmani and in one pool of Lu. longipalpis the Leishmania species was not determined due to the low quality of the sequences.”

change like:

“In seven pools of Ny. whitmani and in one pool of Lu. longipalpis positive for leishmania DNA, the parasite species was not determined due to the low quality of the sequences.”

We agree with your comment. It is possible that are more than one monoxenous trypanosomatids within our molecular detection. However, based on the small fragment (~400 pb) it is not of the sequence it is not wise to assume this fact. We hope in the future we are able to isolate these trypanosomatids and confirm their identity as species level. We include the phrase in results "These five non-Leishmania sequences found in our study, Herpetomonas spp. and Crithidia spp., are sufficiently different between them and possibly represent two and three, respectively, different MOTUs (Molecular Operational Taxonomic Units)." In the discussion we include “Possibly, more than one species within each genera were detected, however, due to the small length of the amplicon it was not possible to confirm it. On the other hand, the molecular detections presented here, may represented different populations (different MOTUs), reinforcing the importance of further studies on these trypanosomatids.”

We changed that phrase in the abstract “In seven pools of Ny. whitmani and in one pool of Lu. longipalpis positive for leishmania DNA, the parasite species was not determined due to the low quality of the sequences.”

Reviewer #3: I find the way the authors responded acceptable and consider the paper much improved. I would still have some issues with the length of acknowledgements (theoretically you can have it as long as the discussion, there is no official limit, but it is simply not good), the way the gel is stained is irrelevant - it was stained somehow and that's all that matters, but I can live with the present version.

We thank the reviewer for the comments

---

## [Editor Report · Decision Letter 3]

21 May 2020

PONE-D-20-05836R3

Diversity of phlebotomine sand flies and molecular detection of trypanosomatids in Brumadinho, Minas Gerais, Brazil

PLOS ONE

Dear Dr. Andrade Filho,

Thank you for submitting your manuscript to PLOS ONE. After careful consideration, we feel that it has merit but does not fully meet PLOS ONE’s publication criteria as it currently stands. Therefore, we invite you to submit a revised version of the manuscript that addresses the points raised during the review process.

We look forward to receiving your revised manuscript.

Kind regards,

Vyacheslav Yurchenko

Academic Editor

PLOS ONE

Additional Editor Comments (if provided):

Thanks for addressing all the comments.

A few final notes on formalities:

1) Make sure all species and generic names are in Italic (they are not in References)

2) Why all words in some titles in References are capitalized (f.e. 44, 52, and others)? Please fix.

3) Please check diacritics.

I am ready to accept the manuscript when these are corrected.

---

## [Author Response · Author response to Decision Letter 3]

22 May 2020

Manuscript ID: PONE-D-20-05836R1

Dear Dr. Yurchenko,

We have reviewed all of your requests.

All references have been fixed. Now, the generic and species names are in italic; we have also removed the capitalized words in some reference titles and diacritics have been reviewed.

We are confident that the new version of the manuscript is greatly improved. 

Looking forward to hearing from you soon.

José Dilermando Andrade Filho

---

## [Editor Report · Decision Letter 4]

27 May 2020

Diversity of phlebotomine sand flies and molecular detection of trypanosomatids in Brumadinho, Minas Gerais, Brazil

PONE-D-20-05836R4

Dear Dr. Andrade Filho,

We are pleased to inform you that your manuscript has been judged scientifically suitable for publication and will be formally accepted for publication once it complies with all outstanding technical requirements.

With kind regards,

Vyacheslav Yurchenko

Academic Editor

PLOS ONE
---

## [Editor Report · Acceptance letter]

9 Jun 2020

PONE-D-20-05836R4 

Diversity of phlebotomine sand flies and molecular detection of trypanosomatids in Brumadinho, Minas Gerais, Brazil 

Dear Dr. Andrade-Filho:

I'm pleased to inform you that your manuscript has been deemed suitable for publication in PLOS ONE. Congratulations! Your manuscript is now with our production department. 

Kind regards, 

on behalf of

Dr. Vyacheslav Yurchenko 

Academic Editor

PLOS ONE